# LEARNING OBJECT AFFORDANCE WITH CONTACT AND GRASP GENERATION

## ABSTRACT

Understanding object affordance can help in designing better and more robust robotic grasping. Existing work in the computer vision community formulates the object affordance understanding as a grasping pose generation problem, which treats the problem as a black box by learning a mapping between objects and the distributions of possible grasping poses for the objects. On the other hand, in the robotics community, estimating object affordance represented by contact maps is of the most importance as localizing the positions of the possible affordance can help the planning of grasping actions. In this paper, we propose to formulate the object affordance understanding as both contacts and grasp poses generation. we factorize the learning task into two sequential stages, rather than the black-box strategy: (1) we first reason the contact maps by allowing multi-modal contact generation; (2) assuming that grasping poses are fully constrained given contact maps, we learn a one-to-one mapping from the contact maps to the grasping poses. Further, we propose a penetration-aware partial optimization from the intermediate contacts. It combines local and global optimization for the refinement of the partial poses of the generated grasps exhibiting penetration. Extensive validations on two public datasets show our method outperforms state-of-the-art methods regarding grasp generation on various metrics.

## 1 INTRODUCTION

Affordance is an area studying how an object can be used by an agent. Understanding affordance can help to design better and more robust robotic systems operating in complex and dynamic environments (Hassanin et al., 2021). For example, a cup can be grasped and passed over by a hand and a bed can be sat or slept onto by a human. Learning affordance (or affordance understanding) has wide applications like grasping (Bohg et al., 2013), action recognition and prediction (Jain et al., 2016; Koppula et al., 2013; Koppula & Saxena, 2015), functionality understanding (Grabner et al., 2011), social scene understanding (Chuang et al., 2018) etc. In this paper, we focus on object affordance for hands, i.e. hand-object interactions.

Though of great importance to many applications, only several works about 3D grasp synthesis using deep learning (Corona et al., 2020; Taheri et al., 2020; Jiang et al., 2021; Karunratanakul et al., 2020; Zhang et al., 2021; Taheri et al., 2021) have been proposed in the computer vision community. In (Taheri et al., 2020), a dataset for human grasping objects with annotations of full body meshes and objects meshes have been collected, and a coarse-to-fine hand pose generation network based on a conditional autoencoder (CVAE) is proposed. In (Karunratanakul et al., 2020), a new implicit representation is proposed for hand and object interactions. The previous work (Taheri et al., 2021) takes a step further to learn dynamic grasping sequences including the motion of the whole body given an object, instead of static grasping poses. Both these work defines affordance as possible grasping poses allowed by the objects. However, instantiations of affordance understanding can include affordance categorization, reasoning, semantic labeling, activity recognition, etc. (Deng et al., 2021)

Among all these, semantic labeling of contact areas between agents and objects is found to be of the most importance (Deng et al., 2021; Roy & Todorovic, 2016; Zhu et al., 2015) because localizing the position of possible affordance can greatly help the planning of actions for robotic hands (Mo et al., 2021; Wu et al., 2021; Mandikal & Grauman, 2021; 2022). In the robotics community, Mo

et al. (2021) and Wu et al. (2021) first estimate the contact points for parallel-jaw grippers and plan paths to grasp the target objects. For dexterous robotic hand grasping, recent works (Mandikal & Grauman, 2021; 2022) find that leveraging contact areas from human grasp can improve the grasping success rate significantly in a reinforcement learning framework. However, they assume an object only has one grasp contact area and learn a one-to-one mapping from an object to the contact.

To overcome the limitation of work in both computer vision and robotics community, we propose to formulate the object affordance understanding as both contacts and grasp poses generation. Specifically, we factorize the learning task into two sequential stages, rather than taking a black-box hand pose generative network that directly learns an object to the possible grasping poses in previous work. 1) In the first stage, we generate multiple hypotheses of the grasping contact areas, represented by binary 3D segmentation maps. 2) In the second stage, we learn a one-to-one mapping from the contact to the grasping pose by assuming the grasping pose is fully constrained given a contact map. Different from a coarse-to-fine strategy, our decomposition not only provides intermediate semantic contact maps, but also benefits from the intermediate task learning in the quality of the generated poses. This intermediate task learning has been proven effective in many computer vision tasks (Tang et al., 2019; Wan et al., 2018; Tome et al., 2017; Wu et al., 2017). In the robotic grasping, it is shown that optimizing grasping poses directly from contacts is superior to re-targeting observed grasps to the target hands (Brahmbhatt et al., 2019b), which also motivates our choice.

Therefore, the other benefit of the intermediate contact representation is enabling the optimization from the contacts. Different from the optimization for the full grasps from scratch in (Brahmbhatt et al., 2019b), we propose a penetration-aware partial optimization from the intermediate contacts. It combines of a local and global optimization for the refinement of the partial poses of the generated grasps exhibiting penetration. The local-global optimization constrains gradients to affect only on the partial poses requiring adjustment, which results in faster convergence and better grasp quality than a global optimization.

In summary, our key contributions are (1) we formulate object affordance understanding as contact and grasp pose synthesis; (2) we develop a novel two-stage affordance learning framework that first generates contact maps and then predicts the grasp pose constrained by the maps; (3) we propose a penetration-aware partial optimization from the intermediate contacts for the grasp refinement; (4) benefiting from the first two decomposed learning stages and partial optimization, our method outperforms existing methods both quantitatively and qualitatively.

## 2 RELATED WORKS

**Grasp Generation** Human grasp generation is a challenging task due to higher degrees of freedom of human hands and the requirement of the generated hands to interact with objects in a physically reasonable manner. Most methods use models such as MANO (Romero et al., 2017) to parameterize hand poses, aiming to directly learn a latent conditional distribution of the hand parameters given objects via large datasets. The distribution is usually learned by generation network models such as Conditional Variational Auto-Encoder (Sohn et al., 2015), or Adversarial Generative Networks(Arjovsky et al., 2017). To get finer poses, many existing works adopt a coarse-to-fine strategy by learning the residuals of the grasping poses in the refinement stage. Corona et al. (2020) uses a generative adversarial network to obtain an initial grasp, and then an extra network to refine it while Taheri et al. (2020) passes hand parameters to a CVAE model and output an initial grasp, followed by a further refinement.

In recent work, however, Jiang et al. (2021) proposes to exploit contact maps to refine human grasps by leveraging the consistency of the contact map. Though estimating the hand-object contact maps, it only reasons about the contact consistency to refine the generated pose while our work exploits the contact maps as an intermediate representation for the final grasp generation. On the other hand, in the area of robotic grasping, Brahmbhatt et al. (2019b) introduces a loss for optimization using contact maps captured from thermal cameras (Brahmbhatt et al., 2019a; 2020) to filter and rank random grasps sampled by Graspit! (Miller & Allen, 2004). It concludes that synthesized grasping poses optimized directly from the contact demonstrate superior quality to other approaches, kinematically re-targeting observed human grasps to the target hand model. The contact maps are also used in the hand and object reconstruction. Grady et al. (2021) proposes a differentiable contact optimization to refine the hand pose reconstructed from an image.

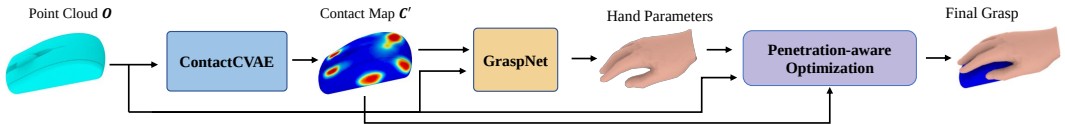

Figure 1: The framework of our method. It consists of three-stages: ContactCVAE, GraspNet and Penetration-aware Partial Optimization. ContactCVAE takes an object point cloud $O$ as input and generates a contact map $C'$. GraspNet estimates a grasp parameterized by $\theta$ from the contact map $C'$. Finally, penetration-aware partial optimization refines $\theta$ to get the final grasp.

**Object Affordance for Robotic Grip** Object affordance is not only used in generating human poses and hand poses, but also important for robotic grippers. Many robotic grasping works focus on parallel-jaw grippers. Yan et al. (2018) learns to rate a proposed gripper pose to be a success or failure in a data-driven way. Mousavian et al. (2019) uses a variational autoencoder(VAE) to generate robotic grasps, and their method can directly work in the real world with only training purely in simulation. In contrast, in (Mo et al., 2021; Wu et al., 2021), locations for grasping points are first predicted and the actions are planned to approach these points. Zhou & Hauser (2017) increases the number of jaws of the gripper and estimate grasp quality using information collected from overhead depth images of novel objects using a modified convolutional neural network (CNN). In recent work for dexterous robotics hands (Mandikal & Grauman, 2021; 2022), contact maps from human grasping are used to guide the planning of the robotic hands based on reinforcement learning.

## 3 METHOD

Figure 1 shows our method pipeline, which generates maps for contact areas by a network naming ContactCVAE, maps the contact maps to grasping poses by the other network naming GraspNet and refines the generated grasp by a penetration-aware optimization module. In the work, we adopt MANO (Romero et al., 2017) to represent grasps. The MANO model $\mathcal{M}$ parameterizes the hand mesh $M = (V, F)$ ($V \in R^{778 \times 3}, F \in R^{1538}$ denotes the mesh vertices and faces) by the shape parameters $\beta \in R^{10}$ and pose parameters $\theta \in R^{51}$, i.e. $M = \mathcal{M}(\theta, \beta)$. In the work, we use the mean shape and use $M = \mathcal{M}(\theta)$ for brevity.

In the first stage, ContactCVAE aims to learn a contact map distribution represented by a latent vector $z$ given an input object by a conditional variational autoencoder. The network takes an object point cloud $O \in R^{N \times 3}$ and the contact map $C \in R^{N \times 1}$ as the input and learns to make the output contact map $C' \in R^{N \times 1}$ as close to the input contact map as possible. $N$ is the number of the points in $O$. Each point in the point cloud is represented by its normalized 3D positions. Each point in the contact map takes a value in $[0, 1]$ representing the contact score. During inference, given an object, a contact map $C'$ can be generated by sampling from $z$. In the second stage, GraspNet learns a mapping from the contact map $C'$ to the hand mesh $M$ constrained by the map. The pose $\theta'$ of the predicted mesh $M'$ from GraspNet is refined with a penetration-aware partial optimization in the third stage.

### 3.1 CONTACTCVAE

**Architecture** Figure 2 demonstrates the architecture of the ContactCVAE network, which is a generative model based on CVAE (Sohn et al., 2015). It consists of two blocks: a Condition-Encoder and a Generator. **Condition-Encoder** The Condition-Encoder $E_{\theta_c}$ is built on PointNet (Qi et al., 2017). It takes a point cloud as input to extract both local features $f_l \in R^{N \times 64}$ and global features, which are duplicated N times to make a feature map $f_g \in R^{N \times 1024}$ for matching the shape of local feature $f_l$. These two features are then concatenated as $f_{lg}$ for the conditional inputs for the generator below.

**Generator Network** The generator $G_{\phi_g}$ follows an encoder-decoder architecture. As shown on the top of Figure 2, the encoder, $E_{\theta_e} : (C, O) \rightarrowtail z$, is based on PointNet (Qi et al., 2017) architecture which takes both an object point cloud $O$ and a contact map $C$ as inputs and outputs the latent code $z \in R^{64}$. The encoder is only employed in training and is discarded in the inference. The latent code $z$ represents a sample of the learned distribution $Q(z|\mu, \sigma^2)$ and is used to generate the contact

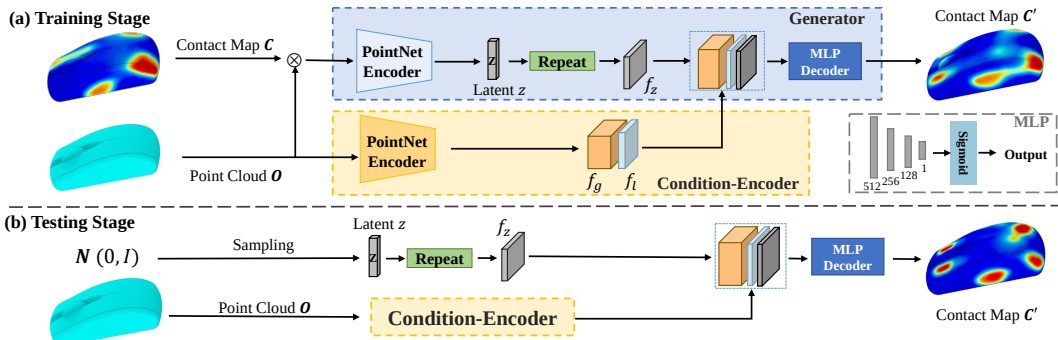

Figure 2: The architecture of ContactCVAE. (a) In the training stage, it takes both an object point cloud and a contact map as input to reconstruct the contact map; (b) In the testing stage, by sampling from the latent distribution, it generates grasp contacts with an object point cloud as the conditional input only. $\otimes$ means concatenation.

map, where $\mu, \sigma^2$ denotes the mean and variance of the distribution. We then duplicate the latent code $z$ N times to make the latent feature $f_z$ for all the points.

The decoder $D_{\theta_d} : (f_z^i, f_{lg}^i) \rightarrowtail C'^i$ is a classifier for a point $i$ which merges three different features (global $f_g^i$, local $f_l^i$ and latent $f_z^i$) to classify whether the point belongs to a contact map or not. The decoder $D_{\theta_d}$ uses the MLP architecture and the weights are shared for all points.

**Testing Stage** During inference, as shown in the bottom of Figure 2, we only employ the Conditional-Encoder and decoder $D_{\theta_d}$. A latent code $z$ is randomly sampled from a Gaussian distribution and forms the latent feature $f_z$. At the same time, the Condition-Encoder takes an object point cloud to output the global and local feature. With these features $(f_z, f_g, f_l)$, $D_{\theta_d}$ outputs the grasp contact $C'$ for the object.

**Contact Loss** The goal of training the model is optimizing $\theta_e, \theta_d$ to reconstruct the contact map well. We simplify the goal as a binary classification task. Thus, we adopt the binary cross-entropy loss for the model over all the points, name as $L_{c1}$. However, some samples have small contact regions and it is hard for the model to learn those samples well by simply adopting the BCE loss. To address this problem, we additionally introduce the dice loss (Milletari et al., 2016) to train the model. It can assist the model in paying attention to small target region. In our work, we adopt the dice loss for the same purpose and name as $L_{c2}$. the formulation of the two loss is defined as:

$$L_{c1} = -\sum_{i=0}^{N}[y_i log(\hat{y}_i) + (1-y_i)log(1-\hat{y}_i)] \tag{1}$$

$$L_{c2} = 1 - \frac{2\sum_{i=0}^{N} y_i \hat{y}_i}{\sum_{i=0}^{N} y_i + \sum_{i=0}^{N} \hat{y}_i} \tag{2}$$

where $\hat{y}_i$ and $y_i$ represent the predicted contact and ground truth of a point $i$, respectively.

Following the training of CVAE (Sohn et al., 2015), we use the KL-Divergence loss regularizing the latent distribution to be close to a standard Gaussian distribution. The loss term is named as $L_{kl}$. The overall loss function of the ContactCVAE network, $L_{contact}$, is represented as:

$$L_{contact} = \gamma_0 L_{c1} + \gamma_1 L_{c2} + \gamma_2 L_{kl}, \tag{3}$$

where the $\gamma_0 = 0.5$, $\gamma_1 = 0.5$ and $\gamma_2 = 1e-3$ are constants for balancing the loss terms.

## 3.2 GRASPNET

**Architecture** With the assumption of hands full constrained by a contact map, we adopt a one-to-one mapping function for getting the grasping pose from the generated contact from the first stage. As shown in the Figure 3, the model takes an object point cloud $O$ and its generated (or reconstructed) contact $C'$ as the input to predict the hand mesh for the grasping pose, which is represented by the MANO model (Romero et al., 2017). Specifically, we employ a PointNet (Qi et al., 2017) to extract the feature, and then use a MLP with four hidden layers to regress the MANO parameters. Given the parameters, the MANO model forms a differentiable layer which outputs the hand mesh $M$.

Contact Map $C'$  Hand Parameters $\theta$  Hand Mesh $M$  Contact Map $C''$

PointNet Encoder MLP MANO

Point Cloud $O$

Figure 3: The architecture of GraspNet. It takes the concatenation of the generated (reconstructed) object contact $C'$ and the point cloud $O$ as input to predict the grasp mesh parameterized by MANO.

During the training period, we use both ground truth and reconstructed contact map to train the GraspNet. During inference, we only use the generated contact map to predict the grasp mesh. Both reconstructed and generated contact maps are from the ContactCVAE model in the first stage.

**Grasp Loss** We simply adopt the reconstruction loss ($L_2$ distance) for the predicted vertices, named as $L_v$. We also use the chamfer distance $Lcd$ between hands and objects, and penetration loss $L_{ptr}$ from (Taheri et al., 2020; Jiang et al., 2021) which punishes penetrations between the hand and object. The loss on MANO parameters is divided into two parts. We use the L1 loss for the translation parameter ($\theta'^t \in R^3$) and the geodesic loss (Mahendran et al., 2017) for the pose parameter ($\theta'^p \in R^{48}$), named as $L_t$ and $L_p$ respectively.

**Consistency Loss** Similar to (Jiang et al., 2021), we introduce the contact consistency loss $L_{cst} = \|C' - C''\|^2$. Based on the distance between the object and the grasp mesh $M$, the contact map $C''$ can be inferred. If the grasp mesh $M$ is predicted correctly from the GraspNet, the input contact map $C'$ should be consistent with the contact map $C''$.

The overall loss of GraspNet, $L_{grasp}$, is the weighted sum of all the above loss terms:

$$L_{grasp} = \lambda_v L_v + \lambda_{cd} L_{cd} + \lambda_{ptr} L_{ptr} + \lambda_t L_t + \lambda_p L_p + \lambda_{cst} L_{cst}, \tag{4}$$

where $\lambda_v$=35, $\lambda_{cd}$=20, $\lambda_{ptr}$=5, $\lambda_t$=0.1, $\lambda_p$=0.1 and $\lambda_{cst}$=0.05 denote the corresponding loss weights.

### 3.3 PENETRATION-AWARE PARTIAL OPTIMIZATION

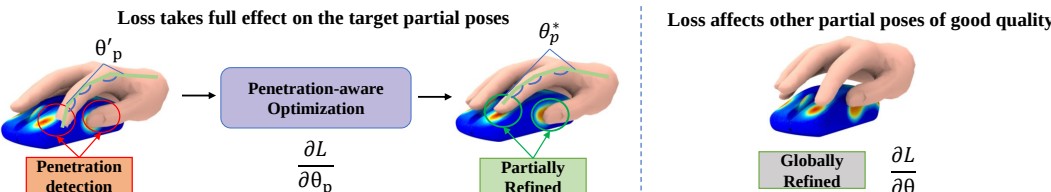

**Loss takes full effect on the target partial poses** $\theta_p^*$ **Loss affects other partial poses of good quality**

$\theta'_p$

Penetration-aware Optimization

$\frac{\partial L}{\partial \theta_p}$

Penetration detection

Partially Refined

Globally Refined $\frac{\partial L}{\partial \theta}$

Figure 4: Left: our penetration-aware partial optimization. Penetration is detected in the index and thumb finger. Partial poses $\theta_p$ for the index finger are pointed out. Right: the result of a global optimization

Though GraspNet gives plausible grasps for most cases, the grasps may exhibit penetration, resulting in low grasp success rate in simulation. Hence, we propose to detect the penetration and refine the partial poses causing it while keep other partial poses of good quality unchanged. The full hand mesh is divided into six parts: five fingers and the palm. If penetration is detected in the palm area, all the poses are adjusted. If penetration is detected in a finger part and no penetration happens in the palm area, only the partial poses of the finger are adjusted. We use $\theta_p$ to represent the partial poses requiring adjustment and the refined poses $\theta_p^*$ are obtained by

$$\theta_p^* = \arg\min_{\theta_p}(\omega_0 L_{cst}(C''(\theta_p), C') + \omega_1 L_{ptr}(\mathcal{M}(\theta_p), O) + \omega_2 L_h(\theta_p, \theta'_p)). \tag{5}$$

$L_{cst}$ is the consistency loss defined above. $C''(\theta_p)$ denotes the process of generating a contact map from the variable $\theta_p$. $L_{ptr}$ penalizes the penetration between the hand and object as similar in (Jiang et al., 2021; Karunratanakul et al., 2020). $L_h$ makes the refined partial poses stay close to the prediction $\theta'_p$ from GraspNet. $\omega_0$=0.01, $\omega_1$=2 and $\omega_2$=0.02.

Figure 4 (Left) shows an example of our partial optimization for the poses $\theta_p$ of the finger. In the refinement stage, as the loss mainly results from local wrong partial poses, the global optimization $\arg\min_\theta L(\theta)$ (Right in Figure 4) has two issues 1) the gradient affects other good poses, 2) the gradient cannot take full effect on the refinement for the wrong partial poses.

## 4 EXPERIMENT

### 4.1 IMPLEMENTATION DETAILS

We sample $N = 2048$ points on an object mesh as the input object point cloud. Our method is trained using a batch size of 32 examples, and an Adam optimizer with a constant learning rate of 1e-4. The training dataset is randomly augmented with $[-1, 1]cm$ translation and rotation at three (XYZ) dimensions. All the experiments were implemented in PyTorch, in which our models ran 130 epochs in a single RTX 3090 GPU with 24GB memory. In the Obman dataset (Hasson et al., 2019), all the ground truth contact map is derived by normalizing the distance between the ground truth of hand and object. For the inference refinement (both partial and global optimization), the Adam optimizer with a learning rate of $2.0 \times 10^{-4}$ is used. In the refinement process, each input is optimized for 200 steps.

### 4.2 DATASETS

**Obman** We first validate our framework on the Obman dataset (Hasson et al., 2019), which is a large-scale synthetic dataset, including 3D hand interacting with objects. The hands are generated by a physical-based optimization engine Graspit! (Miller & Allen, 2004), and are parameterized by the MANO model (Romero et al., 2017). The dataset contain 8 categories of everyday objects selected from ShapeNet (Chang et al., 2015) with a total of 2772 meshes. The model trained on this dataset will benefit from the diversified object models. The object contact map is derived as (Taheri et al., 2020) by thresholding the normalized distance between the object points and their nearest hand vertices. Points with the distance smaller than a threshold are marked as contact.

**ContactPose** The ContactPose dataset (Brahmbhatt et al., 2020) is a real dataset for studying hand-object interaction, which captures both ground-truth thermal contact maps and hand-object poses. Though the dataset contains only 25 household objects and 2306 grasp contacts, it captures more real interactions. For example, the contact in ContactPose spreads across large sections of the hand, as opposed to that at the fingertips for most cases in Obman. We manually split the dataset into a training and test group according to object type. Specifically, we use 4 objects (cup, toothpaste, stapler, and flashlight) with 336 grasp contacts as a test set, and the rest for training the model. ContactPose uses the thermal camera-based method to capture the contact region.

### 4.3 EVALUATION METRICS

A good generated pose should be physically stable and should be in contact with the object without Penetration. In this work, we adopt three metrics to evaluate the quality of generated grasp poses: (1) **Penetration** The penetration is measured by the depth (Dep, $cm$) and the volume (Vol, $cm^3$) between the objects and generated hand meshes. The depth is the maximum or mean of the distances from the hand mesh vertices to the surface of the object if a penetration occurs. Following (Jiang et al., 2021; Karunratanakul et al., 2020), the volume is measured by voxelizing the hand-object mesh with voxel size $0.5cm$. (2) **Simulation Displacement** (Sim-Disp) The simulation displacement is adopted to measure the stability of the generated grasp. We report the average (Mean, $cm$) and variance (Var, $cm$) of the simulation displacement as measured by a physics-based simulator following the same settings as (Jiang et al., 2021; Karunratanakul et al., 2020). The displacement is the Euclidean distance between the object centers before and after applying a grasp on the object. Though used in the existing work, the results of previous work (Karunratanakul et al., 2020) indicate that a high penetration might correspond to a low simulation value and therefore we suggest readers use it for a rough reference only. (3) **Contact Rate** (CR, %) A physically plausible hand-object interaction requires contact between the hand and the object. We define a sample as positive if the hand-object contact exists, which means that there exists at least a point on the hand surface is on or inside the surface of the object. The contact rate is the percentage of those positive samples over all the test samples.

In addition to the metrics used in the hand generation work, we introduce two more metrics to evaluate the quality of the grasping pose distributions. (4) **Grasp Success Rate** (Sim-SR, %) The grasp success rate aims to evaluate the rate of grasp success. Specifically, we define the positive sample as the one with penetration-vol$< 5cm^3$ and simulation-mean $< 2cm$. The success rate is the percentage of those positive samples over all the test samples. (5) **Diversity** (Div, $cm$) It is also

Table 1: Quantitative comparison on Obman test set.

| Methods | Penetration (↓) | | Sim-Disp (↓) | | CR (↑) | Div (↑) | Sim-SR (↑) |
|---|---|---|---|---|---|---|---|
| | Dep | Vol | Mean | Var | | | |
| Grabnet (Taheri et al., 2020) | - | 8.41 | 1.66 | - | 98.25 | 7.93 | 27.60 |
| GraspField (Karunratanakul et al., 2020) | 0.56 | 6.05 | 2.07 | ±2.81 | 89.40 | - | - |
| Jiang et al. (2021) | 0.46 | 5.12 | **1.52** | ±2.29 | 99.97 | - | - |
| Param2Mesh (Baseline) | 0.70 | 13.07 | 1.75 | ±2.88 | 98.56 | 8.31 | 21.63 |
| Ours w/o refine | 0.45 | 5.20 | 1.70 | **±2.22** | **100.00** | 9.20 | 47.21 |
| Ours global refine | 0.50 | 4.15 | 2.00 | ±2.50 | 97.01 | 8.89 | 58.89 |
| Ours | **0.44** | **3.94** | 1.74 | ±2.28 | **100.00** | **10.18** | **61.37** |
| GT | 0.01 | 1.70 | 1.66 | ±2.44 | 100.00 | 7.86 | 87.12 |
| Ours w/o refine (GT) | 0.43 | 4.68 | 1.97 | ±2.40 | 100.00 | 7.89 | 49.67 |
| Ours (GT) | 0.36 | 3.75 | 1.98 | ±2.46 | 99.86 | 8.16 | 57.19 |

Table 2: Quantitative comparison on ContactPose test set.

| Methods | Penetration (↓) | | Sim-Disp (↓) | | CR (↑) | Div (↑) | Sim-SR (↑) |
|---|---|---|---|---|---|---|---|
| | Dep | Vol | Mean | var | | | |
| GrabNet(coarsenet) (Taheri et al., 2020) | 0.76 | 13.42 | 1.10 | ±1.54 | 97.51 | 6.11 | 16.31 |
| GrabNet(refinenet) (Taheri et al., 2020) | 0.92 | 16.74 | **1.04** | ±1.60 | 97.42 | 5.92 | 16.89 |
| Param2Mesh (baseline) | 1.02 | 19.18 | 1.14 | ±1.79 | 81.84 | 6.37 | 13.19 |
| Ours w/o refine | 0.73 | 10.06 | 1.08 | **±1.18** | **98.85** | 7.40 | 24.71 |
| Ours global refine | 0.70 | 7.49 | 1.58 | ±1.79 | 96.23 | 7.56 | 32.07 |
| Ours | **0.68** | **6.09** | 1.23 | ±1.85 | **98.85** | **7.66** | **38.08** |
| GT | 0.45 | 6.64 | 0.73 | ±1.09 | 100.00 | 6.60 | 41.50 |
| Ours w/o refine (GT) | 0.60 | 9.54 | 0.95 | ±0.97 | 100.00 | 6.79 | 33.93 |
| Ours (GT) | 0.51 | 5.91 | 1.06 | ±1.13 | 99.65 | 7.03 | 39.12 |

significant to evaluate the diversity for the generation task. In this work, we use MAE to measure the diversity of generated results. Specifically, we measure the divergence between each generated sample and all other samples and then average them. Referring to the research (Guo et al., 2020), the formulation is defined as Eq. 6.

$$Diversity = \frac{1}{N_g(N_g - 1)} \sum_{i=0}^{N_g} \sum_{k=0}^{N_g} \|v_i - v_k\|_2, \qquad (i \neq k) \qquad (6)$$

$N_g$ is the number of generated samples. $v_i, v_k$ represent the i-th and k-th generated sample.

### 4.4 SELF COMPARISON

To verify the contribution of our proposed factorization, we construct three variants of our method. **Param2Mesh Baseline**: A baseline for grasp generations. It learns the latent distribution of MANO parameter directly, which is similar to the first stage of the previous work (Taheri et al., 2020). Specifically, we use the same CVAE of our Contact2Grasp to make a fair comparison but replace the encoder $E_{\theta_e}$ of our ContactCVAE with fully connected layers to take the MANO parameters as inputs. Given a 3D object point cloud and a random sample for the distribution, the decoder $D_{\theta_d}$ generates MANO parameters directly. Also, the coarsenet (Taheri et al., 2020) can be served as a baseline as well but with a different architecture. **Ours w/o refine** : A variant of our method removing the final testing time refinement. **Ours global refine**: A variant of our method adopting global refinement at the third stage. **Ours (GT)** shows GraspNet in the second stage trained and tested on GT contract maps and **Ours w/o refine (GT)** without the refinement. By comparing to this variant, we are able to verify the quality of our generated contact maps from ContactCVAE.

**Effectiveness of Two Stage Learning Factorization** Results on both Obman (Table 1) and ContactPose datasets (Table 2) show our proposed method can improve Param2Mesh significantly on all metrics, indicating the effectiveness of the two stage factorization. The penetration-vol improve by 60% (Ours w/o refine) and 70% (Ours) on Obman. A reasonable grasp pose should embody both low penetration and simulation displacement. Thus the Sim-SR is a more comprehensive metric considering both of them. It can be observed that our method shows the best Sim-SR result on Obman (61.37%), improving Param2Mesh by 118%.

On ContactPose, we also compare our method with other variants in diversity. Our method generates more diverse samples than the Param2Mesh baseline and the variant using ground truth contact

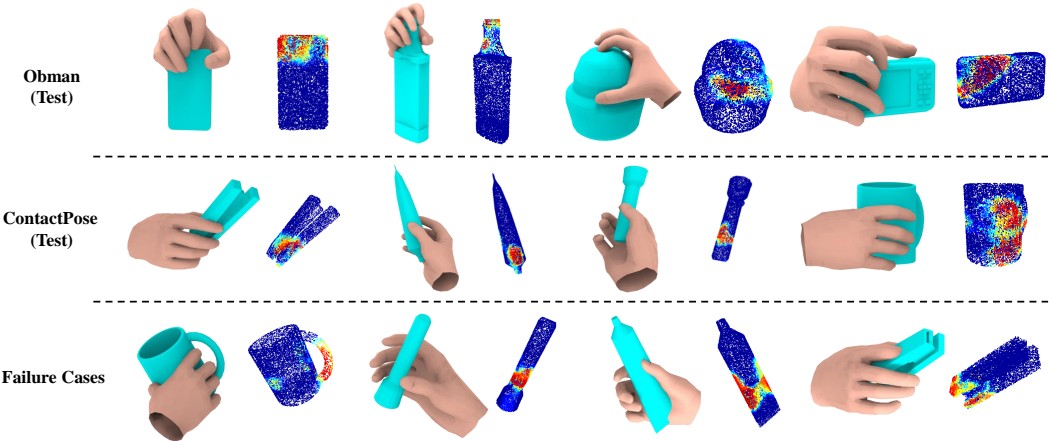

Figure 5: The visualization of generated contact and grasp for objects from Obman test set and ContactPose test set. We also present some failure examples. For each example, we present both the predicted grasp pose (left) and the corresponding contact map (right), which is presented in the form of heat map. Note that all the contact map is generated from our method.

maps. For the latter, as in the second stage only a one-to-one mapping is learned, the diversity actually measures the ground truth poses while our method in the first stage can generate new samples and therefore it is reasonable to have more diverse samples than the ground truth poses.

**Effectiveness of Penetration-aware Partial Optimization** Results on Obman (Table 1) and ContactPose (Table 2) indicate that partial optimization strategy (Ours) achieves better performance than global refinement (Ours glob refine) over all the metrics, which presents the effectiveness of the proposed method.

As shown in Figure 5, we can observe that the generated contact map is reasonable, corresponding to the grasp pose. Although there are some failure examples (including unstable grasps and serious penetration), the hand pose is substantially natural as human behavior.

**Quality of Generated Contact Maps** When compared with Our (GT) and Our w/o refine (GT), the metrics for penetration of our methods are worse but the margin is relative small compared to our improvement from the baseline. For example, on Obman Table 1, penetration depth of our methods are 0.45mm and 0.44mm while those of Our (GT) and Our w/o refine (GT) are 0.43mm and 0.36mm. The comparison indicates the generated maps in the first stage are of high quality. The examples in Figure 5 also show that the generated contact convey meaningful information for grasping.

**Semantic Analysis of Latent Contact Map Space** Using the generated object contacts to formulate the hand grasp is one of contributions and here we show whether our Contact2Grasp can learn the latent distribution for contact well. In the point generation work (Achlioptas et al., 2018), it demonstrates the quality of the generation model by showing that the learned representation is amenable to intuitive and semantically rich operations. Inspired by the work, we conduct the semantic analysis of the latent space learned from our Contact2CVAE model as detailed below.

First, we use ContactCVAE to generate N contact maps and their corresponding latent z. Second, we cluster contact maps using mean shift and adopt the dice metric to measure the distance between different contact maps. Then we select a sample from each cluster set randomly. Each sample consists of a contact map and its corresponding latent z. Then, we interpolate between the latent z of different samples to get a new latent z, and then the new latent z is fed into the decoder of ContactCVAE to get its corresponding contact map. In the paper, we interpolate 2 latent z and therefore get 2 contact maps. Finally, their contact maps are fed into GraspNet to get their corresponding grasp poses.

Figure 6 exemplifies two pairs of the types selected and the contact maps and grasping poses interpolated between them. Notice that the grasps change gradually in alignment with the contact maps between Type A and Type B. For example, in Figure 6 (b), the yellow arrow and circle denote small differences between the contact maps and the grasp poses. As the contact region gradually appears, the middle finger moves to the corresponding position smoothly. Similar interesting observation can

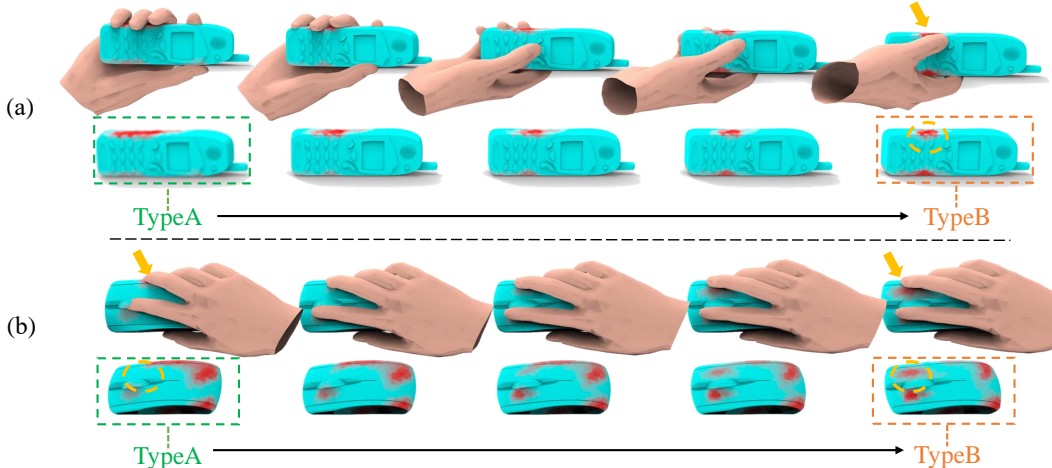

Figure 6: Interpolated contact maps and grasps between different types of generated contacts (TypeA and TypeB). Note that the grasping poses (e.g. finger positions denoted in the yellow circle and arrow) change with transitions between two types of contacts.

be found for manipulating the phone in Figure 6 (a) where the hand poses change from holding to pressing gradually.

## 4.5 COMPARISON WITH EXISTING WORK

When compared with the state-of-the-arts, our method achieves the least penetration, highest grasp success rate, highest contact rate and highest diversity on both Obman and ContactPose. The improvement can be attributed to the factors as follows. Existing work (Taheri et al., 2020; Jiang et al., 2021) aims to learn the mapping relationship between the object and grasp pose distribution directly, e.g. point clouds to joint rotations. Such mapping relationship is highly non-linear. In comparison, our proposal decomposes the hard problem into two tasks by introducing intermediate contact maps. The point clouds of object and the contact areas represented by the segmentation maps on the point clouds are aligned in the same 3D space, which is relatively easy to learn. In ContactCVAE, we also make full use of the alignment and design a local point feature to point classification mapping in the structure.

**Limitations** Although our method achieves significant performance, there exists some limitations. The major limitations of our method can be summarized in two aspects. (1) The two-stage learning: The error from the first stage cannot be corrected. If a generated contact map from the first stage is of low quality, the resulting grasp may be influenced as well. (2) Ambiguity of Contact Map: In some cases, the contact maps are ambiguous, which results in more than one plausible grasping pose. Therefore, the assumption of the one-to-one mapping in the second stage does not hold. To address the limitations mentioned above, some solutions can be explored as the future work. (1) Contact Filter: Before inputting the generated contact maps to the GraspNet, a clustering algorithm (e.g. mean shift) can be adopted to filter those deviating too far from the distribution. (2) More detailed contact maps: To address the ambiguity problem, we can introduce more-detailed contact maps, which contains contact areas with different parts of the hands providing stronger constraints.

## 5 CONCLUSION

In this paper, we propose a novel framework for human grasps generation, which holds the potential for different deep architectures. The highlight of this work is exploiting the object affordance represented by the contact map, to formulate a functionality-oriented grasp pose and using penetration-aware partial optimization to refine partial-penetrated poses without hurting good-quality ones. The proposed method is extensively validated on two public datasets. In terms of diversity and stability, both quantitative and qualitative evaluations support that, our method has clear advantages over other strong competitors in generating high-quality human grasps.

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
