# OpenReview forum: "Learning Object Affordance with Contact and Grasp Generation"
_ICLR.cc/2023/Conference — Submitted to ICLR 2023_

### Official Review · Reviewer_3pXU · 2022-10-20

**Confidence:** 4
**Correctness:** 3
**Technical Novelty And Significance:** 2
**Empirical Novelty And Significance:** 4
**Recommendation:** 3

**Clarity, Quality, Novelty And Reproducibility:**

The paper writing is good, clear, and easy to follow. The work is built upon some prior works addressing the same problem but has a new insight. However, we can also question the novelty of the "new insight" if we look at many grasping works in the robotics community. The paper tries to say that prior works on human-like hand pose prediction for object grasping use end-to-end and one-stage prediction (goes from the object point cloud input to the hand parameters directly), and this paper proposes first to predict where the hand should touch (the "contact map") before predicting the actual hand parameters. If we look at the grasping works using a parallel-jaw gripper in robotics, many works (such as GraspNet) actually only predict how the robot hand (a parallel-jaw gripper) should go (similar to the idea of a contact map in the paper). And how the hand goes there is usually accomplished via motion planning. So I would say there are also some prior works using a similar idea, but in the context of parallel-jaw grasping setups.

**Strength And Weaknesses:**

**Strength**:

* I think the idea of first predicting the contact map before predicting the grasp pose is intuitive and natural. The design choices made in the paper in order to achieve this two-stage prediction are also quite straightforward. For example, to process the point cloud, the paper uses PointNet. To train the contact map prediction network, the paper uses a conditional VAE, as there exist many possible contact maps.

* The paper found that when optimizing the hand pose to avoid penetration, it's better to isolate the optimization on the fingers and on the palm. Local optimization avoids affecting other regions of the hand. This is a potentially useful observation for other robotic hand problems.


**Weakness**:

* The paper has few novel technical contributions other than the idea of first predicting the contact map in order to predict the hand pose. Most of the techniques or the loss function designs used in the paper are from prior works and are quite straightforward (in the sense that other people who come to the problem are very likely to come up with a similar solution). On the one hand, it's good that the pipeline is simple and may be easily reproduced. On the other hand, the paper is not technically novel.

* While the paper presents a pipeline to infer the contact map for different objects and some plausible hand grasp poses, there exists a significant gap to apply such models to a real/simulated robotic setting. For example, the paper considers a grasp pose good if it satisfies constraints such as no/small penetration and small object movement. However, to actually grasp the object with a robot hand, even in simulation, contact physics matters, small prediction errors on the pose can lead to drastically different contact forces and then lead to immediate grasp failure, etc. In addition, the assumption of having access to the complete noise-free point cloud can be a problem in real robotics settings as we typically get a partial view of an object from a camera. More works need to be carried out in order to apply such models to more realistic robotic settings.

* Some of the evaluation metrics are not convincing. For example, the `grasp success rate` is defined by the simulation displacement and the amount of penetration. It is not a good definition for grasp success rate. As mentioned above, low-level physics matters. It's unclear if the prediction satisfies penetration-vol< $5cm^3$ and simulation-mean < 2cm, then it can actually enable a successful grasp. Where do these threshold values come from? Why not just place the object on a table in a physics simulator such as pybullet or mujoco and then actually have the simulated hand execute the predicted grasp pose, and see if the object can be stably grasped (one can check it by lifting the hand and shaking it, and see if the object drops). This would be a proper way to measure the grasp success rate. I would like to see the true grasp success rates in the tables. Similarly, for the definition of `contact rate`, is one contact point between the hand and the object sufficient for a good grasp?  If not, this metric would be less useful.

* The paper does not provide enough details on the evaluation details. As metrics are a critical part of the evaluation, it's better to explain the exact procedures to get the quantitative results. For example, what simulator is used to check the `Simulation Displacement`, and how are the objects and the hand are spawned in the simulator, and how is the displacement measured, how many simulated steps are waited for before measuring the displacement, etc.

* Even though the paper reports that the proposed method outperforms the baselines on the defined metrics, it's unclear how much does such improvements contribute to better performance on a downstream task such as grasping. For example, when the maximum penetration depth is reduced from 0.56 to 0.44cm, how much improvement can one expect to get if the hand executes such poses for grasping the objects? One point I am trying to say is that the connection between the reported metrics and the actual physics simulation/motion is too weak.

**Summary Of The Paper:**

Given the complete point cloud of an object, how can a vision model infer the right grasp pose for a human hand such that the object can be grasped stably? The paper tries to address this issue via a staged pipeline. First, it trains a generative model (conditional VAE) to reconstruct the ground-truth contact map of the input object point cloud. Second, given an object point cloud, we can first use the decoder in cVAE to sample a possible contact map, and then train another network to predict the hand grasp pose conditioned on the predicted contact map. The main difference compared to prior works (end-to-end, single stage) on this line is the two-stage training: first predict the contact map, then predict the grasp pose.

**Summary Of The Review:**

The paper presents an intuitive two-stage prediction pipeline for inferring how a hand can grasp objects and shows improvements over prior works on several metrics. The method is quite simple and straightforward. Considering the prior works on this exact problem as well as many works on parallel-jaw grasping, this paper is more like an incremental work. From a robotics perspective, I would also like to see how the defined metrics connect to the performance of an actual downstream task (such as grasping in a physically realistic simulator). And more details on how the metrics are measured would be necessary. It would make the paper much stronger if the paper can include well-designed grasping tests even just in a simulator.

---

### Official Review · Reviewer_8WmN · 2022-10-24

**Confidence:** 4
**Correctness:** 3
**Technical Novelty And Significance:** 2
**Empirical Novelty And Significance:** 2
**Recommendation:** 5

**Clarity, Quality, Novelty And Reproducibility:**

The illustrations and experimental results are clear to me.
Nevertheless, there are some issues in the experiment to be addressed.


**Strength And Weaknesses:**


Strength:
1.	The proposed approach is technically solid. The subcomponents are mainly based on some existing techniques, such as cVAE and PointNet.
2.	The paper is clearly written, and the illustrations are easy to understand.

Weaknesses:
1.	The proposed approach seems to be a combination of the existing methods (e.g., contact map generation, grasp pose prediction) and lacks technical novelty.
2.	The ablation study should include Ours w/o ContactVAE.
3.	The proposed method does not outperform Jiang et al. significantly. In Penetration(Dep), In CR, Ours(100.00%) only outperforms Jiang et al.(99.97%) for 0.03%. In Sim-Disp(Mean), Ours(1.74) is even worse than Jiang et al.(1.52).
4.	Besides, the comparison between Jiang et al. and Ours seems unfair since Jiang et al. does not use test time refinement. If we compare Ours wo refinement with Jiang et al., Ours wo refinement only achieves slightly better results in Penetration(Dep) and CR.


**Summary Of The Paper:**

The paper touches on an important problem in embodied AI, object affordance understanding. The authors formulate the object affordance in a grasp pose generation paradigm. The grasp pose is predicted by a three-stage pipeline: A contact map is first predicted by a cVAE and then is input to a GraspNet to obtain the hand parameters. The final grasp pose is obtained by Penetration-aware Optimization over the hand parameters.

**Summary Of The Review:**

Overall, the proposed method is technically solid, though it lacks technical novelty and does not outperform Jiang et al. significantly.  I would consider increasing my rating if the authors could address my concerns mentioned in the weaknesses section.

---

### Official Review · Reviewer_CDdw · 2022-10-31

**Confidence:** 3
**Correctness:** 3
**Technical Novelty And Significance:** 2
**Empirical Novelty And Significance:** 2
**Recommendation:** 6

**Clarity, Quality, Novelty And Reproducibility:**

### Clarity
* Most of the things are explained in detail, however sometimes it becomes really hard to follow the point and understand it. For eg. `Param2Mesh Baseline'needs to be rewritten and there are grammatical mistakes as well.

### Quality
* Authors have performed extensive empirical experiments, ablation studies, comparison with SOTA baseline(their own version), interpolation of grasps in latent space to show the efficacy of their approach. Only concern I have is that the writing of the paper needs to be improved in order to make it more readable.

### Novelty
* I feel its incremental addition to the current approach, where they have divided the single stage approach into 2 steps and have achieved better results.

### Reproducibility
* All the details about network, data and learning parameters are mentioned in the paper to reproduce the results.


**Strength And Weaknesses:**

### Strength
* Decomposed the problem into 2 stages and have shown its effectiveness over one stage approach.
* Carried out extensive experiments to show the efficacy of approach.
* Shown the effectiveness of latent space to interpolate intermediate grasps.

### Weakness
* Writing can be improved a lot. Sometimes it becomes really hard to follow the paper.
* One-to-one mapping from affordance segmentation to hand joint angles might not be true in many cases. This is also being mentioned in the limitation of paper.
* In Table 1 & 2, it looks like `Ours` has better results than `Ours(GT)`, which is very counter intuitive. In the paper it is mentioned that generated maps produced by the proposed approach are of high quality which might be the reason behind it. It would be great to see some visualization of GT vs predicted affordance mask to see the difference.
* In figure 5, the last row, first column (for hand holding coffee mug), predicted pose and corresponding contact map does not match. It feels that something is off there.



**Summary Of The Paper:**

In this paper, authors proposed a new framework for human grasps generation for holding objects in hand. Instead of directly learning mapping from object point cloud to joint angles of hand, authors divide this whole problem into 2 steps. In the first stage, authors have tried to learn ContactCVAE, which given point cloud representation of an object and random latent vector, tries to figure out contact segmentation masks for objects. In the second stage, based on the generated contact segmentation mask, joint angles of hands are being learned using GraspNet. The results of GraspNet are further optimized based on penetration aware partial optimization. Authors have shown that this approach yields better results than single stage approach. To support their findings authors have shown both quantitative and qualitative results on Obman & ContactPose dataset.

**Summary Of The Review:**

I have provided `marginally above the acceptance threshold` rating to the paper. In this paper authors have divided 1 stage approach into 2 stage approach which have shown improvement. However I feel, novelty wise it's an incremental improvement. To show the efficacy, authors have carried out extensive empirical experiments. Also writing wise paper needs good amount improvement to make it more readable.

---

### Official Review · Reviewer_7dNk · 2022-11-03

**Confidence:** 3
**Correctness:** 3
**Technical Novelty And Significance:** 2
**Empirical Novelty And Significance:** 2
**Recommendation:** 3

**Clarity, Quality, Novelty And Reproducibility:**

These comments are in addition to those above.

The predicted contact map from the ContactCVAE is essentially defining the affordance of the object and subsequent hand grasp. However, the grasp may depend on the task being performed - for opening a bottle the grasp should be around the cap of the bottle, while for drinking from a bottle the grasp should be around the main cylindrical body of the bottle. Is the current framework able to support this task-dependent affordance? If not, how do the authors envision handling this in the long term?

It is not immediately clear from the paper, but is the ContactCVAE and GraspNet jointly optimized? The entire framework seems to be differentiable, so I could imagine jointly optimizing for the losses to potentially alleviate issues with the ContactCVAE generating erroneous contact maps and give more robust grasps.

In Section 4.3, am I correct in saying the diversity metric measures the diversity of the sampled contact maps and hence grasps for the objects? If so, why is it desirable to have a high diversity and how is this beneficial for downstream tasks?

For the “Ours (GT)” and “Ours w/o refine (GT)” approach in Section 4.4, the paper says that it shows GraspNet in the second stage trained and tested on GT contact. Does this mean that the ContactCVAE stage is completely skipped in this case and we directly go to partial optimization after GraspNet?

**General Writing Fixes:**
- Top of Page 3, typo for Object Affordance for Robotic **Grippers**, not Grip.
- Use math mode for “N” times, e.g. in Section 3.1.
- $Lcd$ typing should be $L_{cd}$ in Grasp Loss subsection in Section 3.2.
- The $L_h$ loss in Section 3.3 is not defined.
- Contact2Grasp and Contact2CVAE is not defined in Section 4.4.
- GT **contact** maps not contract maps in Section 4.4.


**Strength And Weaknesses:**

**Strengths**
- The proposed framework is intuitive and easy to follow. Breaking down the problem of generating hand grasps to first predicting contact maps is a sensible approach and is representative of an affordance-based representation.
- The penetration-aware optimization is a straightforward yet effective approach for refining and correcting any errors in the grasp predicted by the GraspNet.
- The experiments demonstrate that the proposed approach results in higher quality grasps in comparison to current SOTA and ablated baselines.
- The qualitative analysis of the interpolation between the latents sampled from the ContactCVAE is interesting and demonstrates that the latent space has incorporated useful information.

**Weaknesses**
- Jiang et al. 2021 also use a CVAE but directly predict the hand parameters. In addition, they also learn a ContactNet to predict the contact map given the hand and object point cloud and then refine the predicted grasp using optimization. The proposed framework feels quite similar to Jiang et al.’s approach, with the main contribution being the reparametrization of the problem as learning to predict intermediate contact maps and a more constrained grasp refinement step. The results demonstrate incremental improvement over this baseline. In addition, why was Jiang et al.’s approach not included in Table 2?
- The writing quality of the paper could be improved and made more consistent. Some arguments are not particularly well-supported. For example:
    - It is arguable that existing work treats object affordance understanding as a “black box”. The models and losses are designed to have the right inductive biases to ensure that learning is amenable. Perhaps it is more appropriate to call approaches that learn the direct mapping from object representations to grasping poses as “end-to-end”.
    - The abstract and introduction makes it seems like there is a distinct difference in affordance prediction between the computer vision and robotics community. This is not particularly true, as there is a large body of work in robotics that directly predicts grasps and forgoes precise contact mapping.
- The coefficients and terms in the loss functions could be justified better. As it stands, they seem to rely on the fact they were used in previous work.
    - For example, the reconstruction loss for the predicted vertices, chamfer distance and penetration dominate the loss for GraspNet. Since this is the case, how important of a role do the other loss terms play?
- In the experimental section, it is unclear how well the framework generalizes to out-of-distribution objects or categories. For the ContactPose dataset, 4 out of the 25 objects are used for testing while the rest for training. It’s not clear to me how that is actually reflected in the metrics or how different those objects are from the training set (e.g. grasping a hammer by the handle is similar to how you might grasp a flashlight by its handle).


**Summary Of The Paper:**

This paper proposes a three-stage approach for generating grasps for a MANO human hand model given a point cloud of an object of interest. This is achieved by modeling object affordances as the task of predicting contact maps and then finding hand parameters that best achieve the desired contacts.

Firstly, a ContactCVAE is used to sample a contact map given the object point cloud. The ContactCVAE labels each point in the point cloud with a contact score to get the contact map, and is trained to generate the ground truth contact map input during training. Secondly, the predicted contact map along with the object point cloud is fed into a GraspNet to predict the parameters for the MANO hand model. As the hand configuration is very high dimensional (63 dimensional), there may be errors in the prediction which lead to penetration of the fingers or the palm into the object. Thus, the authors propose to use penetration-aware partial optimization in the final stage of the proposed approach to refine the hand parameters predicted by the GraspNet.

Through experiments on two datasets, the authors demonstrate that their approach is able to outperform baselines on various metrics including penetration, simulation displacement, and contact rate. Additionally, ablations show that the partial optimization is important for improving performance.

**Summary Of The Review:**

The paper presents a sensible solution to the problem of determining grasping poses by firstly predicting contact maps and then optimizing a grasp pose for the desired contacts. This is in contrast to existing work which directly predicts a grasp pose and then attempts to refine it. The different components of the proposed framework are well-motivated and easy to follow.

Although the results demonstrate improved performance over existing work, the overall architecture of the framework is not particularly novel. In addition, several details of the paper remain unclear which should be addressed including the diversity metric and generalization capabilities.

---

### Decision · Program_Chairs · 2023-01-20

**Decision:**

Reject

**Justification For Why Not Higher Score:**

The novelty is marginal and the empirical results unconvincing.
There was no rebuttal.

**Justification For Why Not Lower Score:**

N/A

**Metareview: Summary, Strengths And Weaknesses:**

This paper proposes a method for generating grasps for a complex human-like hand on novel objects represented as point clouds.  It works in three stages, first finding plausible contacts, then generating grasp configurations for the hand, then adjusting to avoid interpenetration.

A strength of the paper is that it takes on a difficult problem of grasping when the hand and objects are complex.

There are weaknesses in the paper in terms of:
- justifying the two-phase approach relative to other "one-phase" approaches that go directly from point cloud to grasp configuration
- empirical comparisons to the closest previous work (Jiang et al 21), which are missing in some cases, and not convincingly better in others
- empirical evaluation of the success of a grasp, which might be important to test in a good physical simulation

Also, in general, the term "affordance" is quite general, meaning all the different kinds of actions or operations you could do to an object (for example, a sponge "affords" squeezing or a patio door "affords" sliding).  This paper is studying a very particular type of affordance, which is grasping.  It might be clearer if the title and introduction made clear that the only type of affordance you are focused on is grasping.